# Development of Piezoelectric Energy Harvester System through Optimizing Multiple Structural Parameters

**DOI:** 10.3390/s21082876

**Published:** 2021-04-20

**Authors:** Hailu Yang, Ya Wei, Weidong Zhang, Yibo Ai, Zhoujing Ye, Linbing Wang

**Affiliations:** 1National Centre for Materials Service Safety, University of Science & Technology Beijing, Beijing 100083, China; yanghailu@ustb.edu.cn (H.Y.); zwd@ustb.edu.cn (W.Z.); ybai@ustb.edu.cn (Y.A.); yezhoujing@ustb.edu.cn (Z.Y.); 2Research and Development Center of Transport Industry of New Materials, Technologies Application for Highway Construction and Maintenance, Beijing 100088, China; 3Department of Civil Engineering, Tsinghua University, Beijing 100083, China; yawei@tsinghua.edu.cn; 4Innovation Group of Marine Engineering Materials and Corrosion Control, Southern Marine Science and Engineering Guangdong Laboratory, Zhuhai 519080, China; 5Joint USTB Virginia Tech Lab Multifunctional Materials, USTB, Department Civil & Environmental Engineering, Virginia Tech, Blacksburg, VA 24061, USA

**Keywords:** piezoelectric ceramic, stacked structure, energy harvesting, power generation, pavement power generation

## Abstract

Road power generation technology is of significance for constructing smart roads. With a high electromechanical conversion rate and high bearing capacity, the stack piezoelectric transducer is one of the most used structures in road energy harvesting to convert mechanical energy into electrical energy. To further improve the energy generation efficiency of this type of piezoelectric energy harvester (PEH), this study theoretically and experimentally investigated the influences of connection mode, number of stack layers, ratio of height to cross-sectional area and number of units on the power generation performance. Two types of PEHs were designed and verified using a laboratory accelerated pavement testing system. The findings of this study can guide the structural optimization of PEHs to meet different purposes of sensing or energy harvesting.

## 1. Introduction

Many new materials and technologies are developed and implemented for road transportation infrastructures [1,2]. The functions of roads are now not limited to carrying traffic loads but expanded to environmental protection [3], energy saving and information collection [4,5]. In recent years, the concept of intelligent infrastructure has been developing rapidly [6]. Pavement energy harvesting technology has become an important part of intelligent pavement systems [7].

Energy harvesting technologies in roads mainly include solar energy harvesting, thermoelectric generators, geothermal energy harvesting and piezoelectric energy harvesting [8,9]. When tires interact with the road surface, a large amount of mechanical energy is generated in the road, which is ultimately dissipated into heat and wasted. If this energy can be collected and utilized, it can be used for various purposes. In view of this, the application of piezoelectric energy harvesting technology in road engineering has been explored [10,11,12]. Piezoelectric transducer structures include the stack type [13,14], cymbal type [15], cantilever type [16,17] and flexible type [11]. Some of these structures have been tested in the laboratory and finite element analysis has been used to analyze the dynamic response of piezoelectric pavements [17,18].

The vibration of automobiles is transmitted to the road surface to generate vibration energy, which can be harvested by piezoelectric cantilever beams. The low energy density of pavement vibration requires the coordination of the cantilever structures’ natural frequency and the environmental vibration frequency, which is difficult to realize under actual working conditions [19]. The strain energy density due to the compressed road material is related to the position of the automobile tire; it can be harvested by a stack piezoelectric structure. Axle load magnitude and vehicle speed affect power generation of stack piezoelectric transducers [20]. In this research, the piezoelectric energy harvester (PEH) harvesting pavement strain energy is taken as the research object, and is shown in Figure 1. The PEH is embedded in the road, the bottom surface is constrained by the pavement, and the side is flexibly connected with the road. The tire load acts on the upper surface of the PEH and the load is transferred to the piezoelectric units through the upper plate of the PEH. The work of the automobile tire on PEH is converted into electric energy via piezoelectric units.

The piezoelectric materials should be designed as piezoelectric energy harvesters (PEHs) with a certain structure and specific protective packaging in order to improve the power generation efficiency and bear the heavy traffic loads on an asphalt pavement. In this study, PEHs for roads were designed as a box with multiple groups of piezoelectric units packaged inside, as shown in Figure 2. Each piezoelectric unit is formed by stacking several pieces of piezoelectric ceramics. In order to improve the generation efficiency of the PEH, it is necessary to study the influence of geometric parameters on power generation performance.

This paper presents a study on the factors that affect the power generation characteristics, including the connection mode of the stacked piezoelectric ceramics, the number of stacked pieces, the ratio of height to sectional area and the number of the units in the PEH. In addition, the power generation performance of two types of PEH were verified and compared in an indoor test road loaded with an accelerated pavement testing (APT) device. The results show that the connection mode and the number of stacked pieces have a minor effect on the total generated energy, but have a significant effect on the open circuit voltage and the charge quantity. The open circuit voltage and the converted electric energy can be increased by increasing the ratio of height to sectional area if the piezoelectric material volume stays the same. In a PEH, under the same force, the energy produced will decrease as the number of units increases. The conclusions of this paper have practical engineering significance for improving the piezoelectric power generation efficiency and material utilization of PEHs under road conditions.

## 2. Materials and Methods

### 2.1. Materials

The positive piezoelectric effect is the basis of piezoelectric power generation or sensing. When the physical pressure is applied to the piezoelectric material, the electric dipole moment in the material will become shorter due to compression. At this time, in order to resist this change, the piezoelectric material will generate equal positive and negative charges on the opposite surface of the material to maintain its original state. The positive piezoelectric effect is essentially a process in which mechanical energy is converted into electrical energy. In this study, we selected cylindrical piezoelectric ceramics, which were polarized along the axial direction. The piezoelectric element has a large piezoelectric coefficient of *d*_33_ and a high electromechanical coupling coefficient of *k*_33_ under the action of axial force.

The material used in this study was lead zirconate titanate piezoelectric (PZT) ceramics. The type of piezoelectric ceramics was PZT-5H, which was fabricated by Baoding Hongsheng Acoustic Equipment Co., Ltd. in Baoding, China. The main parameters of the material are shown in Table 1.

### 2.2. Multilayer Stack Structure

When the cylindrical piezoelectric ceramics are polarized along the axial direction, the positive electrode will induce a negative charge and the negative electrode will induce a positive charge under the axial compressive force. A single ceramic disk cannot be made very thick due to the limitation of production technology. In order to increase the electrical energy generated, multiple pieces of ceramic disks need to be stacked. According to the polarity of adjacent surfaces of ceramic parts, the connection modes include a parallel connection and a series connection (as shown in Figure 3). For the series stack structure, increasing the number of layers of piezoelectric ceramics is equivalent to increasing the thickness of the piezoelectric ceramics.

### 2.3. Theoretical Analysis

A piezoelectric unit is a capacitive device. The formula of electrical energy stored by a capacitor is the following:(1)$$E=\frac{1}{2}QU$$
where *Q* is the charge quantity, in C; and *U* is the open circuit voltage, in V.

According to the piezoelectric effect equation, the calculation formula of the generated charge *Q*_0_ and the open circuit voltage *U*_0_ on the polarized surface of a one-piece piezoelectric ceramic is the following:(2)$$Q_{0}=d_{33}T_{33}A_{0}=d_{33}F$$
(3)$$U_{0}=\frac{d_{33}Fh_{0}}{\varepsilon_{r}\varepsilon_{0}A_{0}}$$
where *d*_33_ is the piezoelectric strain constant, in pC/N; *T*_33_ is the vertical compressive stress, in Pa; *A*_0_ is cross sectional area of disks and *A*_0_ = π*r*^2^, in m^2^; and *F* is the vertical force and *F* = *T*_33_
*A*_0_, in N. *ε**_r_* is the relative dielectric constant; *ε*_0_ is the vacuum dielectric constant, 8.854187817 × 10^−12^ F/m; and *h*_0_ is the thickness of single piece of ceramic, in m.

According to the formula of capacitance energy storage, the electrical energy *E*_0_ produced by a single piezoelectric ceramic disk under the load *F* can be calculated as follows:(4)$$E_{0}=\frac{1}{2}Q_{0}U_{0}=\frac{1}{2}\frac{d_{33}^{2}F^{2}h_{0}}{\varepsilon_{r}\varepsilon_{0}A_{0}}$$

According to Equations (2)–(4), the open circuit voltage and induced charge of *n*-piece stack structure piezoelectric ceramics can be deduced.

When *n* pieces of piezoelectric ceramics are stacked in a parallel connection, under the external load *F*, the open circuit voltage *U**_p_*, the generated charge *Q_p_* and electrical energy *E_p_* are respectively as following:(5)$$Q_{p}=nd_{33}F$$
(6)$$U_{p}=\frac{d_{33}Fh_{0}}{\varepsilon_{r}\varepsilon_{0}A_{0}}=\frac{d_{33}Fh_{t}}{n\;\varepsilon_{r}\varepsilon_{0}A_{0}}$$
(7)$$E_{p}=\frac{1}{2}\frac{nd_{33}^{2}F^{2}h_{0}}{\varepsilon_{r}\varepsilon_{0}A_{0}}=\frac{1}{2}\frac{d_{33}^{2}F^{2}h_{t}}{\varepsilon_{r}\varepsilon_{0}A_{0}}$$
where *n* is the number of piezoelectric ceramic pieces in a piezoelectric stack structure; *h**_t_* is the total height of the stack piezoelectric structure, m; and *h**_t_* = *nh*_0_.

When *n* pieces of piezoelectric ceramics are stacked in a series connection, under the external load *F*, the open circuit voltage *U_s_*, the generated charge *Q_s_* and the electrical energy *E_s_* are respectively as follows:(8)$$Q_{s}=d_{33}F$$
(9)$$U_{s}=\frac{nd_{33}Fh_{0}}{\varepsilon_{r}\varepsilon_{0}A_{0}}=\frac{d_{33}Fh_{t}}{\varepsilon_{r}\varepsilon_{0}A_{0}}$$
(10)$$E_{s}=\frac{1}{2}\frac{nd_{33}^{2}F^{2}h_{0}}{\varepsilon_{r}\varepsilon_{0}A_{0}}=\frac{1}{2}\frac{d_{33}^{2}F^{2}h_{t}}{\varepsilon_{r}\varepsilon_{0}A_{0}}$$

The stack structures can be composed of several ceramic pieces in series and parallel. According to the above equations, the parallel connection will affect the amount of charge, but will not affect the open circuit voltage; and the series connection is just the opposite. They generate the same amount of electricity power. When the total height of the stack piezoelectric units is fixed, the number of layers affects the parallel mode. The open circuit voltage is inversely proportional to the number of layers, and the induced charge is directly proportional to the number of layers. However, the total amount of energy is not affected in the above cases. If the total volume of piezoelectric units is fixed, with the increase of the height to cross-sectional area ratio, the voltage and power generation both increase, and the charge is not affected.

### 2.4. Experimental Method

#### 2.4.1. Piezoelectric Unit Test

In this experiment, the test system included loading equipment and the electrical signal acquisition devices (as shown in Figure 4). The loading equipment was the Hydraulic Servo Universal Material Testing Machine-Cooper HYD 25-II. It can provide a maximum load of 25 kN, a load frequency range of 0 to 70 Hz, and a temperature range between −20 and +60 °C (±0.2 °C). Its load control accuracy is 0.1 kN, and its frequency control accuracy is 0.1 Hz. In this study, the loading equipment HYD 25-II was used to provide sinusoidal load. The loading parameters to be set include load frequency, *f*, load amplitude, *F**_amp_*, and mean load, *F**_mean_*. The force applied to the piezoelectric body is two times that of the *F**_amp_* with a preload, which is equal to *F**_mean_* minus *F**_amp_*. The load-time equation is as follows:(11)$$F\left(t\right)=F_{amp}\sin\left(2\pi ft\right)+F_{mean}$$
where *F**_amp_* is the load amplitude, in kN; *f* is the load frequency, in Hz; and *F**_mean_* is the mean load, in kN. In the experiment, an oscilloscope and a charge amplifier were used to obtain the voltage signal and the charge signal. The oscilloscope can directly measure the voltage waveform of analog signal. The type of the oscilloscope was Tektronix MDO 3014 with four analog channels and 16 digital channels. Its bandwidth is 100 MHz and its sampling rate is 2.5 GS/s. The recording length is 10 M points. The high voltage probe P5100a was selected, which can measure 2.5 kV peak voltage. It can meet the measurement range of the high output voltage of piezoelectric ceramics.

The charge amplifier ICA 105 was used for charge measurement. The charge amplifier module converts the amount of charge into the voltage value, which can be measured with an oscilloscope. The charge amplifier sensitivity is 10^−5^ C/V. The charge amplifier can set the magnification to 1, 2, 10, 25 and 200 times and the maximum measurement range is ±3 × 10^−5^ C.

The electrical properties of piezoelectric units are mainly characterized by open voltage and charge quantity. The open circuit voltage *U* is measured with the oscilloscope. The generated charge *Q* is measured with the charge amplifier and the oscilloscope. As a capacitive electronic component, the electrical energy generated in the piezoelectric structure can be calculated using the formula of capacitance energy storage. In this research, the loading waveform is sinusoidal, and so are the wave form of the produced voltage and charge. Under continuous sinusoidal load, the maximum and minimum values of voltage signal will drift, and the peak–peak value will remain unchanged. We compared the peak–peak value of the open circuit voltage *U*_p-p_ and the peak–peak value of the generated charge *Q*_p-p_ to analyze the influence of connection mode, stack layer number and height/diameter ratio on power generation performance.

#### 2.4.2. PEH Test

To investigate the influence of the number of the piezoelectric units on the power generation of a PEH, we designed a PEH that can install 19 units at most, as shown in Figure 5. The built-in piezoelectric units are all connected in parallel. The upper and lower stainless steel plates are used as electrodes. The upper and lower protective packaging is tightened by bolts. The load acts on the upper surface of the outer package, and the stress is transferred to the stainless steel plate through the upper plate of the package structure, and then distributed to each piezoelectric unit.

Different numbers of piezoelectric units ranging from eight to 15 were placed in the PEH and the sinusoidal load was applied by the Cooper HYD 25-II machine. The oscilloscope and charge amplifier were used to monitor the voltage and charge signals, and the influence of the number of units on the power generation performance was analyzed.

#### 2.4.3. Laboratory Pavement Loading Test

The laboratory test system consisted of an indoor testing road, an accelerated pavement testing (APT) device, an energy collection module, and an oscilloscope, as shown in Figure 6. The test road was 2.8 m in length, 0.9 m in width and 24 cm in thickness. A circular PEH and a square PEH were embedded in the indoor test road to form the indoor piezoelectric test road. The circular PEH was 30 cm in diameter and 10 cm in height, with 19 groups of piezoelectric units inside it. The square PEH was 30 cm in length, 30 cm in width and 6.7 cm in height, with nine groups of piezoelectric units inside it. The PEHs were embedded in the asphalt road with their upper surface leveled with the road surface.

The APT machine is an MMLS-1/3 (PaveTesting, Hertfordshire, UK), which was used to supply the moving wheel loading. It has five loading wheels and can apply high-frequency loading. The maximum load frequency was 7200 times per hour. The maximum wheel load was 2.7 kN; the tire diameter was 30.0 cm, and the maximum wheel pressure was 0.8 MPa.

The energy collection module has an energy collection circuit based on an LTC 3588 chip and a 1 F super capacitor, which was used to store the electric energy. LTC 3588 is a commercial energy harvesting chip developed by Linear Technology Corporation. It is especially designed for piezoelectric energy collection. The LTC3588 integrates a low-loss full-wave bridge rectifier with a high efficiency buck converter to form a complete energy harvesting solution optimized for high output impedance energy sources. The schematic and printed circuit board (PCB) of the piezoelectric energy collection circuit are shown in the Figure 7. The output voltage can be set to 1.8, 2.5, 3.3 or 3.6 V by the D0 and D1 select. In this test, the output voltage was set to 3.6 V.

The oscilloscope was Tektronix MDO 2024 with four analog channels, which was used to monitor the voltage of the energy storage capacitor.

## 3. Determination of Structural Parameters of Piezoelectric Unit

In order to optimize the structure parameters of the piezoelectric unit, including the connection mode, the number of stacking layers and the height to sectional area ratio, three tests were designed and conducted.

### 3.1. Test A: Determination of Piezoelectric Unit Connection Mode

In this test, we designed and fabricated two specimens labeled A-1 and A-2, which were both stacked with three pieces of piezoelectric ceramics PZT-5H, as shown in Figure 8. A-1 was in parallel connection mode and A-2 was in series connection mode. Each PZT-5H piece’s size was Φ20.00 × 7.50 mm and the overall size was Φ20.00 × 22.50 mm. The geometric parameters and loading scheme are shown in Table 2 and Table 3.

The open voltage peak–peak value *U*_p-p_, the charge peak–peak value *Q*_p-p_ and the converted energy E of A-1 and A-2 are shown in Figure 9.

As seen in Figure 9, in the multilayer stack piezoelectric structure, the series connection voltage was higher and the charge quantity was lower as compared to the parallel connection. The generated electrical energy of specimen A-1 was more than that of A-2, which is not completely in accordance with Equations (7) and (10). This may be due to the differences in the material piezoelectric strain constant *d*_33_, the relative dielectric constant *ε**_r_* and the accuracy of the measurement method. The charge quantity of A-1 was about three times that of A-2, which is close to the theoretical value. However, the open circuit voltage of A-1 was not about one third that of A-2. The series stack piezoelectric structure has a very high voltage and low current, which is not suitable for power extraction. On the premise that the total height and cross-sectional area of the piezoelectric unit remain unchanged, the parallel structure can optimize the voltage and current by adjusting the number of parallel layers, which cannot be realized by the series structure.

### 3.2. Test B: Determination of Number of Piezoelectric Unit Stacking Layers for Parrallel Connection Mode

The voltage and charge are affected by the number of layers of parallel stack structure. In this test, three specimens were designed, fabricated and labeled B-1, B-2 and B-3, which all had a multilayer parallel connection, as shown in Figure 10. The geometric parameters and loading scheme are shown in Table 4 and Table 5.

The open voltage peak–peak value *U*_p-p_, the charge peak–peak value *Q*_p-p_ and the converted energy E of B-1, B-2 and B-3 are shown in Figure 11.

As shown in Figure 11, in the multilayer stack piezoelectric structure of parallel connection, as the number of layers increased, the open circuit voltage decreased and the charge increased. The ratio of charge quantity *Q*_B-1_:*Q*_B-2_:*Q*_B-3_ was close to 3:4:5 and the ratio of open voltage *U*_B-1_:*U*_B-2_:*U*_B-3_ was close to 1/3:1/4:1/5, as shown in Table 6.

The differences in the power generation may be caused by the differences of the piezoelectric coefficient *d*_33_ and the relative dielectric constant *ε**_r_*, which cannot be guaranteed to be identical in the same batch of piezoelectric materials. The parallel structure can optimize the power extraction by increasing the number of layers. By adjusting the number of layers in the parallel stack structure, the energy can be optimized, but the total power generation cannot be significantly improved.

### 3.3. Test C: Determination of Height to Cross-Sectional Area Ratio of Stacking Structure

In this test, three specimens were labeled as C-1, C-2 and C-3. C-1, which were all in series connection mode, as shown in Figure 12. The geometric parameters and loading scheme are shown in Table 7 and Table 8.

The open voltage peak–peak value *U*_p-p_, the charge peak–peak value *Q*_p-p_ and the generated energy E of C-1, C-2 and C-3 are shown in Figure 13.

As shown in Figure 13, for cylindrical piezoelectric structures, the larger the height to cross-sectional area ratio was, the higher the open circuit voltage was, and the more electrical energy was generated. However, this had little effect on the amount of charge. If we define the ration of height to the cross-sectional area as parameter $a=h_{t}/A_{0}$, Equations (9) and (10) can be written as follows:(12)$$U_{s}=\frac{ad_{33}}{\varepsilon_{r}\varepsilon_{0}}$$
(13)$$E_{s}=\frac{1}{2}\frac{ad_{33}^{2}F^{2}}{\varepsilon_{r}\varepsilon_{0}}$$
where a is the ratio of total height *h**_t_* of the stack piezoelectric structure and cross-sectional area *A*_0_, and $a=h_{t}/A_{0}$.

According to Equations (12) and (13), the open voltage *U**_s_* and generated electrical energy *E_s_* are proportional to a. In Test C, the ratios of the total height to the sectional area of C-1, C-2 and C-3 were $a_{C-1}=0.161$,$\;a_{C-2}=0.064$ and $a_{C-3}=0.025$ and $a_{C-1}:a_{C-2}:a_{C-3}=6.44:2.56:1$. From the results of Test C shown in Figure 10, The ratio of the open circuit voltage *U*_C-1_:*U*_C-2_:*U*_C-3_ and the ratio of the generated electrical energy *E*_C-1_:*E*_C-2_:*E*_C-3_ are shown in Table 9. The order of voltage and electrical energy of the three specimens was consistent with the theoretical results. The difference between the numerical ratio and the theoretical ratio is mainly due to the fact that the piezoelectric coefficient *d*_33_ and the relative permittivity *ε**_r_* of the material may not be absolutely the same and the accuracy of ultra-high voltage measurement was limited, which will also affect the measurement results. Even so, it can be clearly seen from the experimental results that the higher the ratio of height to diameter was, the higher was the voltage, and the more electric energy was generated.

## 4. Determination of the Number of the Piezoelectric Units in One PHE

As shown in Figure 2 and Figure 5, PEH has piezoelectric units arranged in an array. The vertical compressive load on the PEH acts on the piezoelectric units to generate electric energy. When all piezoelectric units are connected in parallel, the arrangement will not affect the power generation performance of the PEH. As shown in Figure 1, the PEH embedded in the road bears the pressure from the tire. The load acts on the upper surface of the outer package, and the stress is transferred to the stainless steel plate through the upper plate of the package structure, and then distributed to each piezoelectric unit. The sum of the loads on each piezoelectric element is equal to the external load. When the external applied load remains unchanged, with the increase of the number of units, the force on each element decreases. On the premise that the structural parameters of the piezoelectric unit have been determined, the number of piezoelectric units will affect the life and power generation performance of the PEH. This part of the research focuses on the influence of the number of piezoelectric units on the power generation performance. The PEH can install 19 units at most, as shown in Figure 5. The piezoelectric unit was made up of three pieces of PZT-5H sized Φ20.00 × 7.5 mm in parallel connection mode. The positive and negative electrodes of each unit were connected by a stainless steel plate. Therefore, all units were connected in parallel.

In this test, the number of piezoelectric units increased from eight to 15, increasing by one each time as shown in Figure 14. The Cooper HYD 25-II applied a sinusoidal load with an amplitude of 5 kN and a frequency of 10 Hz. The oscilloscope and charge amplifier were used to monitor the voltage peak–peak value *U*_p-p_ and charge peak–peak value *Q*_p-p_. The test results are shown in Figure 15.

As shown in Figure 15, with the increasing number of piezoelectric units, the voltage decreased, the charge quantity was not affected and the total energy decreased. If the PEH in Figure 5 is regarded as one piezoelectric unit, the change of the number of units can be equivalent to the change of the cross-sectional area of the piezoelectric unit. According to the Equations (5)–(7), it can be concluded that the charge has nothing to do with the cross-sectional area, the voltage decreases with the increase of the cross-sectional area, and the generated energy also decreases with the increase of the cross-sectional area. The test results are in agreement with the theory.

Thus, on the premise of ensuring the structural stability and bearing the traffic load, the number of piezoelectric units should be reduced as much as possible (ensuring its fatigue life) to improve the power generation performance.

## 5. Power Generation Performance Test of PEH in Laboratory Pavement under Moving Load

According to the method described in Section 2.4.3, the indoor laboratory pavement was built and the power generation was tested as shown in Figure 14. Wheel loads were applied to the PEH and the voltage of the energy storage capacitor was recorded every 1000 wheel loads. Square PEH and circular PEH were tested. The units of the square PEH and the circular PEH were made up of three pieces of PZT-5H sized Φ20.00 × 7.5 mm in parallel connection mode, which is the same as the units described in Section 4. The square PEH had nine units and the circular PEH had 19 units. Figure 16a shows the small APT device MMLS 1/3 working on the laboratory test road. The wheel load was 2.6 kN and the tire pressure was 0.7 MPa. The loading frequency was 3444 times per hour and wheel speed was 4.5 km/h. Figure 16b shows the laboratory test road, which had one circular PEH and one square PEH embedded in it. Inside the circular PEH there were 19 piezoelectric units and inside the square PEH there were nine. Figure 16c shows the monitor system including an oscilloscope used to monitor the voltage of the supercapacitor and a laptop used to extract and store the oscilloscope waveform screenshots. Figure 16d shows the waveform screenshots of the voltage, which show a pulse voltage signal.

Figure 17 shows the voltage of the supercapacitor per 1000 wheel loads. The rising trend of voltage met the expectation. According to the formula of the storage energy of the super capacitor, the generated electrical energy can be calculated. The voltage increased from 0.084 to 3.36 V after wheel loads for the square PEH, the stored energy was 5.64 J, and the average generated electrical energy was 0.564 mJ per load repetition. The voltage increased from 0.09 to 3.1 V for the circular PEH, the stored energy was 4.80 J, and the average generated electrical energy was 0.48 mJ per load repetition. The square PEH generated more energy than the circular PEH. The circular PEH had more piezoelectric units than the square PEH, which is consistent with the conclusion of Section 4.

## 6. Conclusions

PZT is the most widely used piezoelectric material and a stack structure is the most suitable for road strain energy harvesting. However, compared with road engineering materials, piezoelectric ceramics have the characteristics of high cost, high stiffness and high modulus. PZT has poor compatibility with pavement materials, so it is not suitable for direct application in roads. It needs a certain structure design and packaging protection to be one PEH and is then applied to the road. In order to optimize the power generation efficiency and improve material utilization, a PEH was designed for road engineering. The stack structure parameters and the number of internal piezoelectric units in the PEH were analyzed. The results are as follows:(1)Under the action of automobile axle load, the columnar piezoelectric unit produced pulse electrical energy with high voltage and very low current, which is unfavorable for energy collection and utilization. The results show that the multi-layer stacked parallel structure should be adopted. When the total height is constant, increasing the number of layers and decreasing the thickness of the layer will not increase the total energy, but it can play a role of reducing the voltage and increasing the current, so as to facilitate the collection and utilization of electric energy.(2)For one piezoelectric unit, when the total volume of the piezoelectric ceramic and the load is constant, the charge does not change with the ratio of height to cross-sectional area. However, the voltage and the generated electric energy is increased with the increasing of the ratio of the height to the cross-sectional area, which means that the piezoelectric unit should be designed slenderer to obtain more electric energy, so as to improve the piezoelectric material utilization.(3)For one PEH with all units connected in parallel mode, by reducing the number of piezoelectric units, the energy collection efficiency is improved. At the same time, the high compressive strength of the PZT is fully utilized to improve the utilization rate of materials.

The research content of this paper has practical significance for the application of stack piezoelectric energy collection technology in road engineering. There are still some problems to be further studied. The range of vehicle loads is very large and road traffic presents the characteristics of less heavy trucks and more light cars. It is needed to study how to optimize the structure in order to generate more power at the actual transportation situation which has a small number of heavy vehicles and a large number of light vehicles. Another topic that has to be studied is how to develop the structural stability of PEHs by optimizing the arrangement of piezoelectric units in the future, in addition, the high efficiency energy extraction circuits that fit the pulse electrical energy should also be improved.

## Figures and Tables

**Figure 1 sensors-21-02876-f001:**
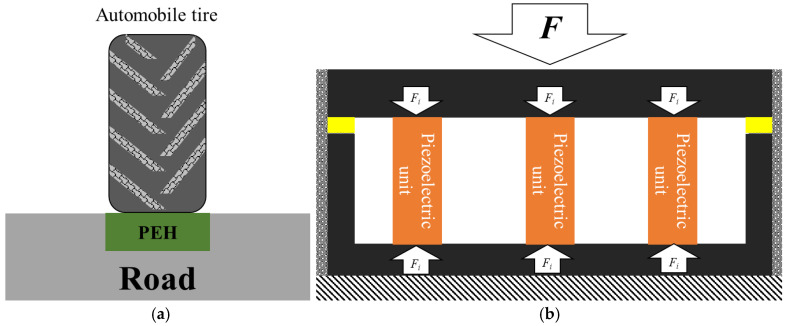
PEH road. (**a**) Schematic of PEH road; (**b**) schematic of internal force of the PEH.

**Figure 2 sensors-21-02876-f002:**
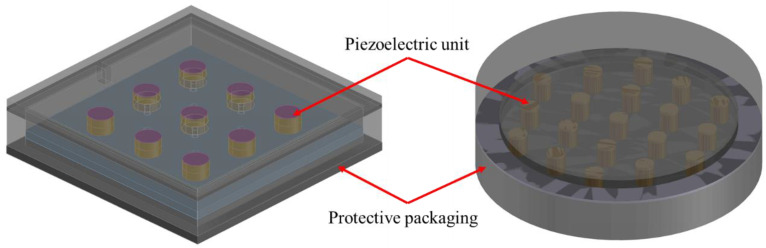
The PEH designed for road energy harvesting.

**Figure 3 sensors-21-02876-f003:**
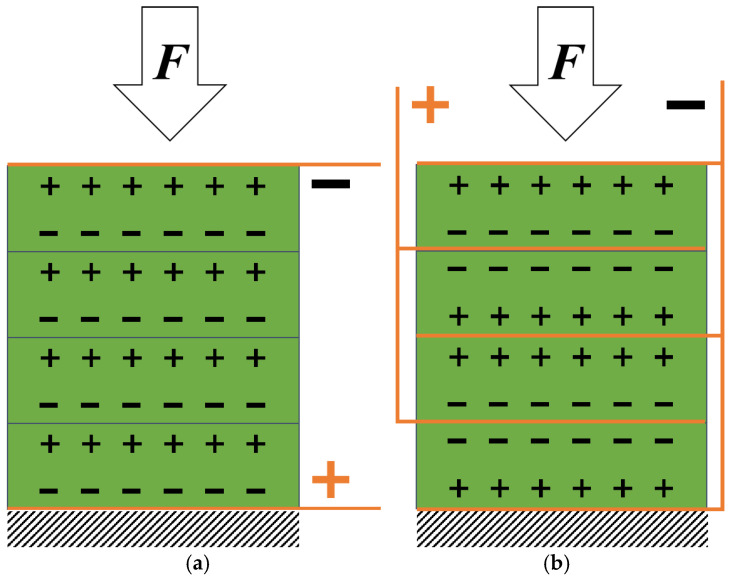
Connection mode of multilayer stacked piezoelectric ceramics. (**a**) Series connection mode; (**b**) parallel connection mode.

**Figure 4 sensors-21-02876-f004:**
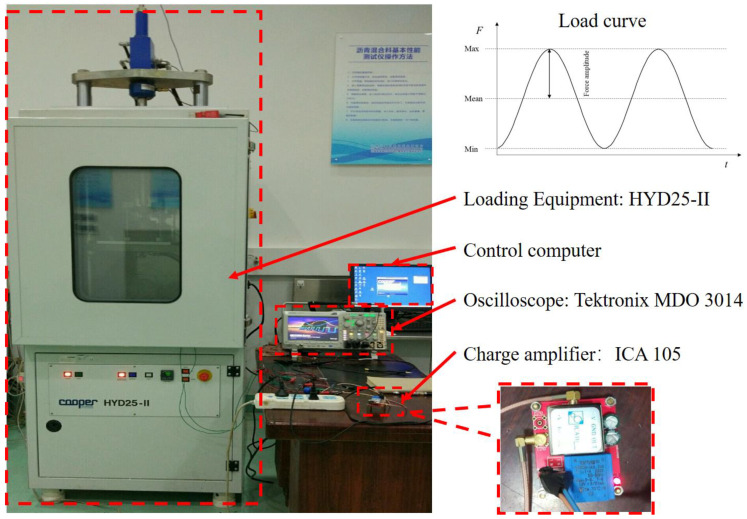
The test system.

**Figure 5 sensors-21-02876-f005:**
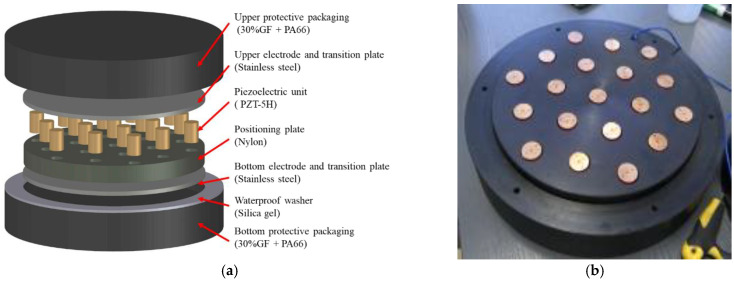
Structure of the PEH. (**a**) Diagram of the PEH. (**b**) Arrangement of piezoelectric units in the PEH.

**Figure 6 sensors-21-02876-f006:**
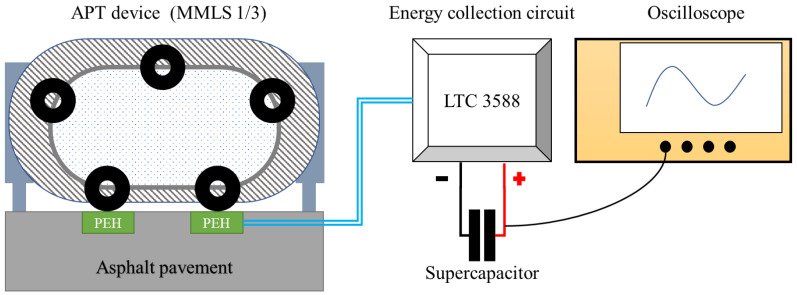
Diagram of the laboratory pavement loading test.

**Figure 7 sensors-21-02876-f007:**
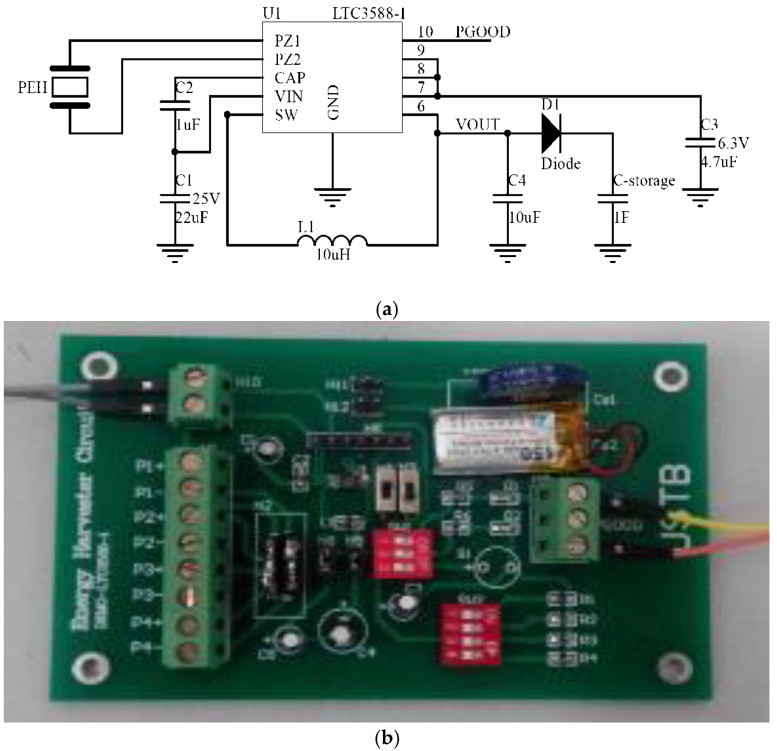
Piezoelectric energy collection circuit. (**a**) Schematic of the circuit. (**b**) The PCB of the circuit.

**Figure 8 sensors-21-02876-f008:**
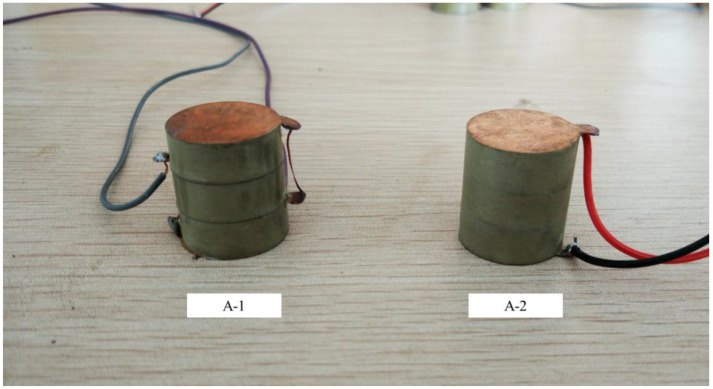
The specimens of Test A.

**Figure 9 sensors-21-02876-f009:**
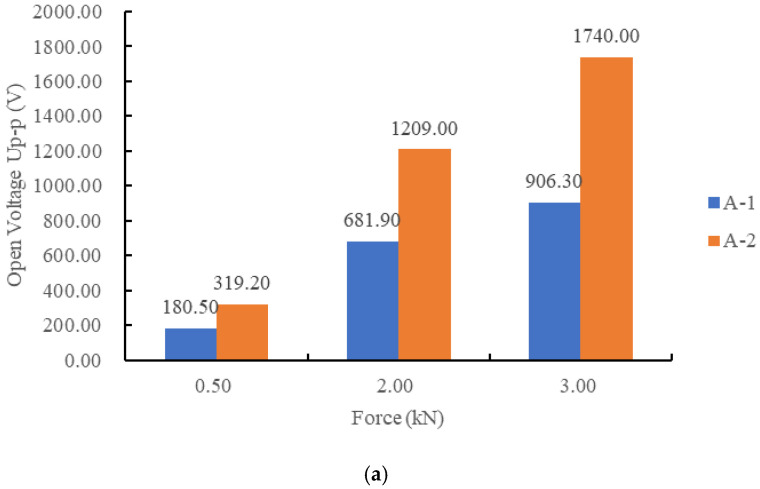
Comparison of electrical properties of different connection modes: (**a**) open peak–peak voltages of A-1 and A-2; (**b**) charge variations of A-1 and A-2; (**c**) generated electrical energy of A-1 and A-2.

**Figure 10 sensors-21-02876-f010:**
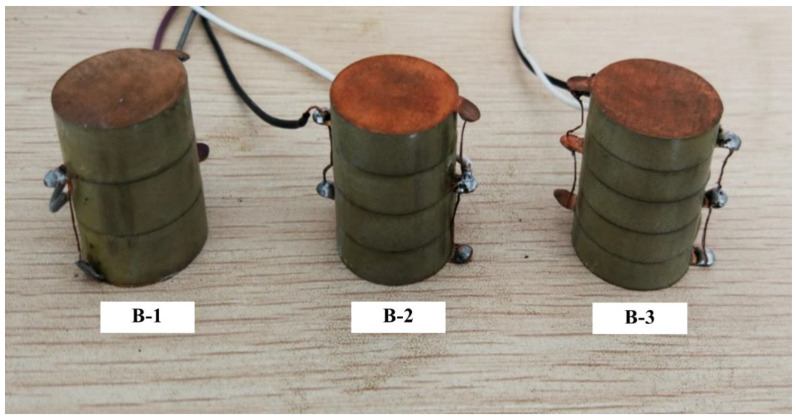
The specimens of Test B.

**Figure 11 sensors-21-02876-f011:**
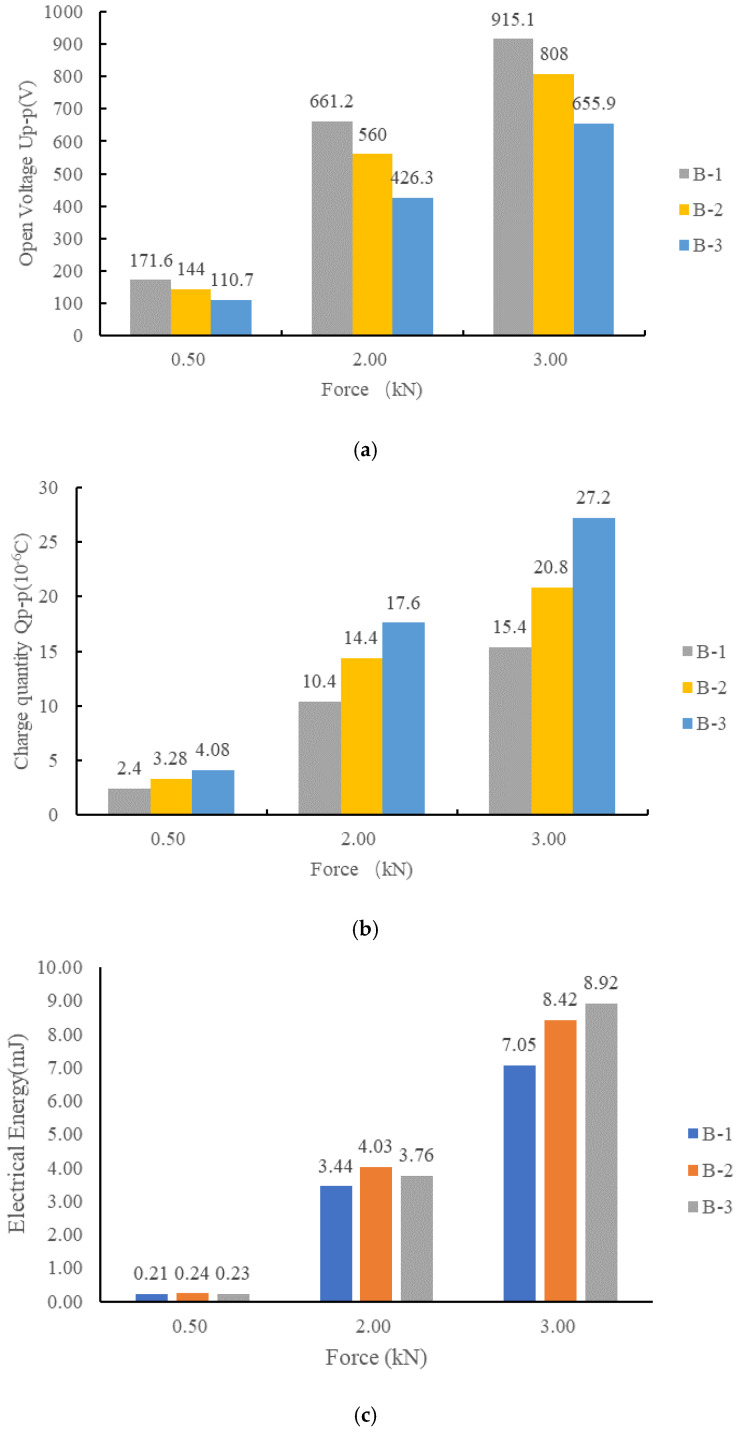
Comparison of electrical properties of stack piezoelectric units with different layers: (**a**) the open peak–peak voltage of B-1, B-2 and B-3; (**b**) charge variation of B-1, B-2 and B-3; (**c**) generated electric energy of B-1, B-2 and B-3.

**Figure 12 sensors-21-02876-f012:**
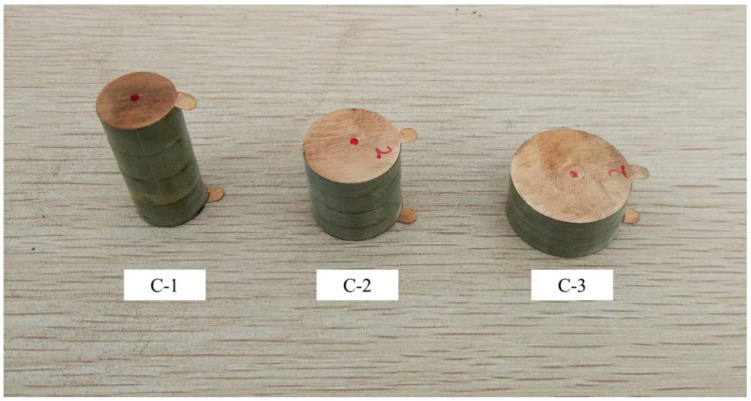
The specimens of test C.

**Figure 13 sensors-21-02876-f013:**
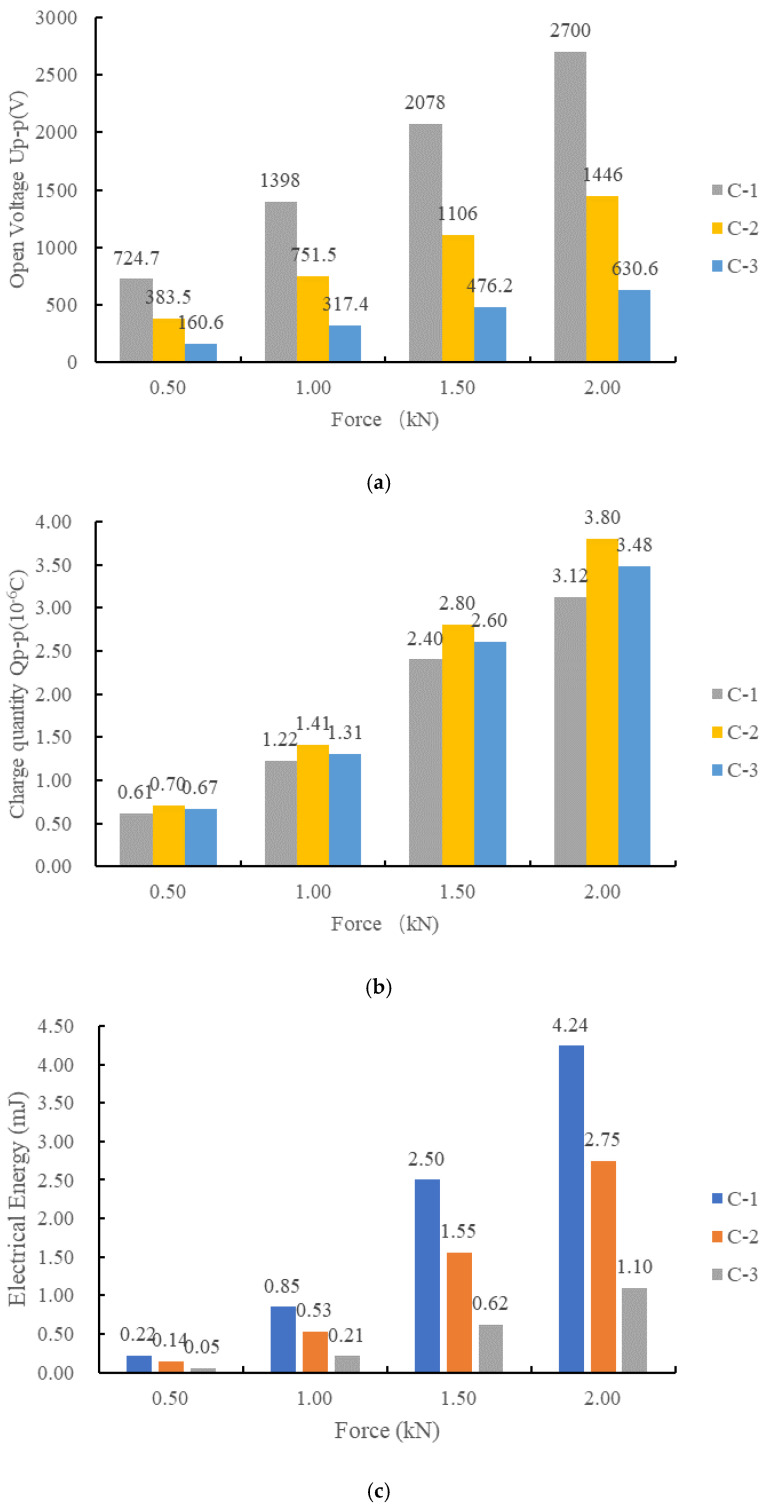
Comparison of electrical properties of stack piezoelectric unites with the same volume and different height to cross section ratio: (**a**) the open peak–peak voltage of C-1, C-2 and C-3; (**b**) charge variation of C-1, C-2 and C-3; (**c**) generated electrical energy of C-1, C-2 and C-3.

**Figure 14 sensors-21-02876-f014:**
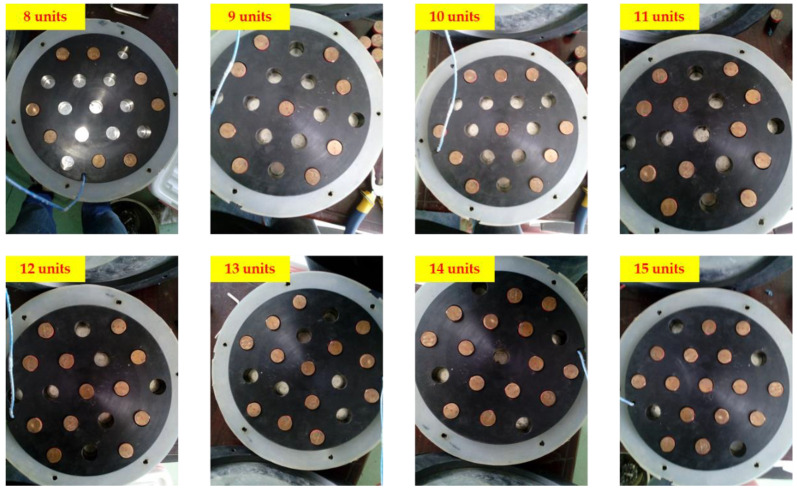
The number of piezoelectric units from 8 to 15.

**Figure 15 sensors-21-02876-f015:**
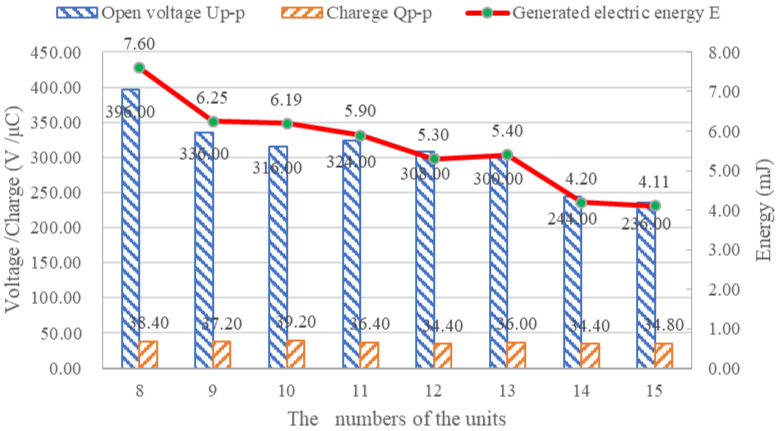
The electric energy signal of the PEH with different numbers of units.

**Figure 16 sensors-21-02876-f016:**
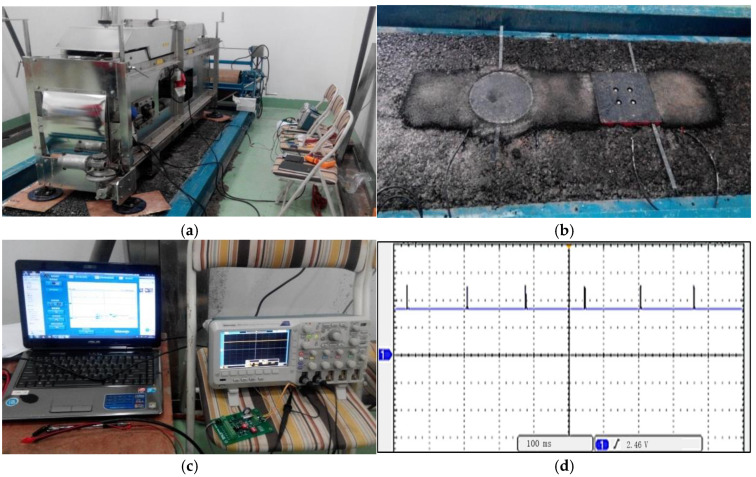
The laboratory pavement loading test of the PEH road. (**a**) The accelerated pavement testing device. (**b**) The laboratory test road. (**c**) Monitor system of the voltage. (**d**) Voltage waveform of the energy storage capacitor.

**Figure 17 sensors-21-02876-f017:**
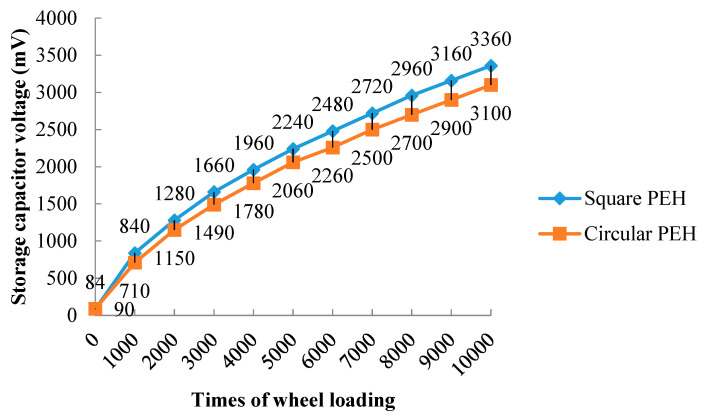
The voltage of the supercapacitor.

**Table 1 sensors-21-02876-t001:** Typical material properties of PZT-5H.

Parameter	Symbol	Unit	Typical Value
Density	*ρ*	10^3^ kg/m^3^	7.45
Curie temperature	*T* _c_	°C	200
Quality factor	*Q* _m_	/	70
Relative permittivity	$\varepsilon_{33}^{T}$	/	4500
Electro-mechanical coupling factor	*k* _33_	/	0.65
Piezoelectric constant	*d* _33_	10^−12^ C/N	670

**Table 2 sensors-21-02876-t002:** The size parameters of the specimens in Test A.

Test	Sample No.	Single Size (mm)	Number of Layers	Total Size (mm)	Connection Mode
A	A-1	Φ20.00 × 7.50	3	Φ20.00 × 22.50	Parallel
	A-2	Φ20.00 × 7.50	3	Φ20.00 × 22.50	Series

**Table 3 sensors-21-02876-t003:** Loading scheme of Test A.

Test	Sample No.	Loading Model	Load Amplitude (kN)	Mean Load (kN)	Frequency (Hz)	Applied Force (Kn)	Preload (kN)
A	A-1, A-2	Sinusoidal load	0.25	0.75	10	0.5	0.5
			1.00	1.50	10	2.0	0.5
			1.50	2.00	10	3.0	0.5

**Table 4 sensors-21-02876-t004:** Size parameters of the specimens in Test B.

Test	Sample No.	Single Size (mm)	Number of Layers	Total Size (mm)	Connection Mode
B	B-1	Φ20.00 × 7.5	3	Φ20.00 × 22.50	parallel
	B-2	Φ20.00 × 5.625	4	Φ20.00 × 22.50	parallel
	B-3	Φ20.00 × 4.50	5	Φ20.00 × 22.50	parallel

**Table 5 sensors-21-02876-t005:** Loading scheme of Test B.

Test	Sample No.	Loading Model	Load Amplitude (kN)	Mean Load (kN)	Frequency (Hz)	Applied Force (kN)	Preload (kN)
B	B-1, B-2, B-3	Sinusoidal load	0.25	0.75	10	0.5	0.5
			1.00	1.50	10	2.0	0.5
			1.50	2.00	10	3.0	0.5

**Table 6 sensors-21-02876-t006:** The proportions of the values of the generated charge and voltage in Test B.

Force (kN)	Ratio of Open Circuit Voltage	Ratio of Charge
	*U*_B-1_:*U*_B-2_:*U*_B-3_	*Q*_B-1_:*Q*_B-2_:*Q*_B-3_
0.50	1/3:1.12/4:1.08/5	3:4.10:5.10
2.00	1/3:1.13/4:1.07/5	3:4.15:5.08
3.00	1/3:1.18/4:1.19/5	3:4.05:5.30

**Table 7 sensors-21-02876-t007:** Size parameters of the specimens in Test C.

Test	Sample No.	Single Size (mm)	Number of Layers	Total Size (mm)	H/CSa Ratio *	Connection Mode
C	C-1	Φ15.87 × 7.94	4	Φ15.87 × 31.76	0.161	series
	C-2	Φ20.00 × 5.00	4	Φ20.00 × 20.00	0.064	series
	C-3	Φ25.20 × 3.15	4	Φ25.20 × 12.60	0.025	series

* H/CSa is the ratio of height to cross-sectional area.

**Table 8 sensors-21-02876-t008:** Loading scheme of Test C.

Test	Sample No.	Loading Model	Load Amplitude (kN)	Mean Load (kN)	Frequency (Hz)	Applied Force (kN)	Preload (kN)
C	C-1, C-2, C-3	Sinusoidal load	0.25	0.75	10	0.5	0.5
			0.5	1	10	1.0	0.5
			0.75	1.25	10	1.5	0.5
			1	1.5	10	2.0	0.5

**Table 9 sensors-21-02876-t009:** The proportions of the voltage and the generated electrical energy in Test C.

Force	Ratio of Open Circuit Voltage	Ratio of Generated Electrical Energy
(kN)	*U*_C-1_:*U*_C-2_:*U*_C-3_	*E*_C-1_:*E*_C-2_:*E*_C-3_
0.50	4.51:2.39:1.00	4.08:2.50:1.00
1.00	4.40:2.37:1.00	4.10:2.55:1.00
1.50	4.36:2.32:1.00	4.03:2.50:1.00
2.00	4.28:2.29:1.00	3.84:2.50:1.00

## Data Availability

Not applicable.

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
