# Peer review of "Development of Piezoelectric Energy Harvester System through Optimizing Multiple Structural Parameters"

_sensors, 2021, doi:10.3390/s21082876_

Round 1
Reviewer 1 Report
This study presents the parameters of the stack piezoelectric transducer on the electrical outputs. My concerns are:
- Page 3, Table 1. The meanings of the symbols should be illustrated.
- Page 8, Table 3. How is the mean load calculated? Is the mean load used in this paper?
- Pages 8-9, Figure 8. How much force is applied to the piezoelectric body? It seems that the forces shown in Figure 8 are different from those given in Table 3. Also check Table 4 and Figure 10, Table 7 and Figure 12.
- Page 15, Section 4. The electrical connection modes of the piezoelectric units should be illustrated in this section. Combine the equations to illustrate the conclusion given in lines 317 -320. Why is the charge quantity not affected by the number of units?
- Page 16, Section 5. The sizes and the electrical connection modes of the piezoelectric units should be illustrated in this section for illustrating the validity of the conclusion in section 4.
Author Response
Thank you very much for your comments. The authors have read your comments carefully, revised or answered them one by one.

Reviewer 2 Report
Title: Development of piezoelectric energy harvester system through optimizing multiple structural parameters
The authors presented research on harvesting electrical energy from pavement using piezoelectric energy harvesters. The topic sounds interesting and their huge efforts were spent on the experiment. However, the manuscript cannot point out their unique points. It is not adequate to be published at the current stage.
The authors should reconsider section 4, pp.15. Why does the generated electric energy decrease when the number of piezoelectric units increases? At the uniform load for each piezoelectric unit, when the number of piezoelectric units increases, the generated energy should increase.
Also, the authors state, in lines 304-308, "... the number and arrangement of piezoelectric units will affect the life and power generation performance of PEH." However, there are only result in the number of piezoelectric units. How about the arrangement with the same number of piezoelectric units? How do you maintain the uniform stress on each piezoelectric unit?
From Fig. 6, your results will be strongly affected by the impedance matching between the harvester and the power management circuit. The efficiency of the power management circuit strongly your final results to make your statement. How do you maintain the same efficiency when changing structural parameters?
In Fig 6 and Fig 15, you show the general diagram and picture of your circuit. However, there is no detail of the electric circuit diagram. This hidden part cannot let your readers self-evaluate your work when they read the manuscript.
For the conclusions (line 354-374), the results (1), (2), and (3) are common in the field. You can find them in other publications. The (4) should be considered again since this is related to how you transfer the load to the harvesters and how uniform it is.
Lines 377 to 384, they might be the unique points of your research and the manuscript should have.
Author Response

(The authors gave the same response as above.)

Reviewer 3 Report
Comments: In this work, the authors designed and verified two types of PEHs through laboratory accelerated pavement testing system to further improve the energy generation efficiency of this type of piezoelectric energy harvester. Finally, the findings of this study can guide the structural optimization of PEH to meet different purposes of sensing or energy harvesting. Considering the novelty and scientific contribution, I am favorable to recommend this paper can be accepted for publication in Sensors. The minor concerns are as follows:
(1) In terms of practical application, energy and power density are two important parameter. What level of energy and power density can be achieved by the piezoelectric unit mentioned in the article?
(2) How far can the energy collected by the piezoelectric unit keep a car running? What is the area of electrodes used? What do you think is the bottleneck that hinders the large-scale application of piezoelectric materials?
(3) What is the range of energy generation efficiency of piezoelectric materials? What is your target value?
Author Response

(The authors gave the same response as above.)

Round 2
Reviewer 2 Report
Please see the attached file for the comments.

Author Response
Thank you very much for your constructive comments. We have carefully revised the content and structure of the article, deleted unnecessary pictures and added background notes. The research contents and conclusions focus on the value of road piezoelectric energy collection. You have offered very professional advice.
